# Identification of HIV-reservoir cells with reduced susceptibility to antibody-dependent immune response

Antonio Astorga-Gamaza, Judith Grau-Expósito, Joaquín Burgos, Jordi Navarro, Adrià Curran, Bibiana Planas, Paula Suanzes, Vicenç Falcó, Meritxell Genescà, Maria J Buzon*

Infectious Disease Department, Hospital Universitari Vall d'Hebron, Institut de Recerca (VHIR), Universitat Autònoma de Barcelona, Barcelona, Spain

**Abstract** Human immunodeficiency virus (HIV) establishes a persistent infection in heterogeneous cell reservoirs, which can be maintained by different mechanisms including cellular proliferation, and represent the main obstacle to curing the infection. The expression of the Fcγ receptor CD32 has been identified as a marker of the active cell reservoirs in people on antiretroviral therapy (ART), but if its expression has any role in conferring advantage for viral persistence is unknown. Here, we report that HIV-infected cells expressing CD32 have reduced susceptibility to natural killer (NK) antibody-dependent cell cytotoxicity (ADCC) by a mechanism compatible with the suboptimal binding of HIV-specific antibodies. Infected CD32 cells have increased proliferative capacity in the presence of immune complexes, and are more resistant to strategies directed to potentiate NK function. Remarkably, reactivation of the latent reservoir from antiretroviral-treated people living with HIV increases the pool of infected CD32 cells, which are largely resistant to the ADCC immune mechanism. Thus, we report the existence of reservoir cells that evade part of the NK immune response through the expression of CD32.

## Editor's evaluation

Persistence of the viral reservoir is hampering HIV cure. This study describes a possible way that HIV-infected cells in the reservoir may escape antibody killing. The findings show that reservoir cells tend to have less of a receptor that binds HIV antibodies capable of cell killing – these cells may then have a survival advantage as they are less susceptible to antibody killing. The study suggests that they also seem to be susceptible to proliferation, which helps maintain the reservoir. These studies provide evidence for one way in which the HIV reservoir is maintained.

## Introduction

HIV establishes a persistent infection for which, nowadays, there is no available cure. Despite huge advances on the optimization of ART, which leads to suppression of viral replication, ART does not fully eliminate the virus from the human body nor can completely solve the persistent inflammation caused by HIV (*Klatt et al., 2013*). Importantly, ART discontinuation leads to viral rebound from diverse anatomical sites and cell subsets containing replication-competent viruses (*Grau-Expósito et al., 2017*; *Jost and Altfeld, 2012*; *Madhavi et al., 2015*) representing the main obstacle in achieving cure (*Joos et al., 2008*).

The HIV reservoir has a complex and heterogeneous nature, where each of the subsets that compose the viral reservoir contributes differently to viral persistence (*Gálvez et al., 2021*; *Astorga-Gamaza*

*For correspondence:
mariajose.buzon@vhir.org

Competing interest: The authors declare that no competing interests exist.

and Buzon, 2021); i.e., central memory cells are one of the main populations contributing to the total reservoir size (Chomont et al., 2009), effector memory cells support HIV transcription (Grau-Expósito et al., 2017) and contain higher proportions of intact viral regions (Duette et al., 2022; Musick et al., 2019; Hiener et al., 2017), and memory stem cells and resident memory T cells are potentially long-lived niches for HIV (Buzon et al., 2014; Cantero-Pérez et al., 2019). Unfortunately, knowledge on the establishment, maintenance, and composition of the reservoir remains incomplete, and the identification of markers to exclusively target persistent HIV-infected cells remains elusive (Darcis et al., 2019; Neidleman et al., 2020). In this regard, the molecule CD32, a low-affinity receptor for the constant fraction of immunoglobulin G (FcγR-IIa), was proposed as a marker of HIV reservoir cells (Descours et al., 2017). While those results were questioned later by the identification of experimental artifacts (Bertagnolli et al., 2018; Pérez et al., 2018), several new studies partially corroborated the original findings; higher levels of viral DNA within the $T_{CD32}$ population were reported after applying a very stringent cell isolation protocol (Darcis et al., 2020) and CD32 was identified as a marker of transcriptionally active persistent HIV-infected cells, both in blood and in the main reservoir tissues, namely the lymph nodes and the gastrointestinal tract (Cantero-Pérez et al., 2019; Abdel-Mohsen et al., 2018; Badia et al., 2018; Vásquez et al., 2019; Noto et al., 2018). Importantly, whether or not CD32 is a marker of latent or transcriptionally active infection, infected CD4$^+$ T cells expressing CD32 contain replication-competent HIV and are found in long-term ART-treated people living with HIV (PLWH) (Cantero-Pérez et al., 2019; Descours et al., 2017; Darcis et al., 2020; Abdel-Mohsen et al., 2018; Badia et al., 2018; Vásquez et al., 2019; Martin et al., 2018). The cell markers CD20 (Serra-Peinado et al., 2019) and CD30 (Hogan et al., 2018) have been shown to also identify transcriptionally active HIV cells in samples from ART-suppressed PLWH. Whether these transcriptionally active HIV-infected cells persist in the body and are not targeted by host immune responses remains unknown.

NK cells are lymphocytes that can eliminate cancer cells or virally-infected cells without prior antigen sensitization. They constitute an important arm of the immune system, not only by a direct cytotoxic effect on aberrant cells but also by the modulation of the adaptive immune responses. NK cells kill target cells by several mechanisms, such as natural cytotoxicity (NC), recognizing stress ligands expressed on the surface of infected cells, or by ADCC, driven by antibodies that bind to target cells (Vivier et al., 2008). The decision of NK cells to kill or not to kill a target cell depends on the balance between activating and inhibitory signals received from the interaction with the target cell (Lanier, 2008). Among relevant NK receptors, we find NC receptors such as NKp46, NKp30, NKp44, different Killer-cell immunoglobulin-like (KIRs), and lectin-like receptors such as NKG2A or NKG2C. Other important activating receptors for NK activity are NKG2D and DNAM-1, whose known ligands are the major histocompatibility complex (MHC) class I-related molecules MICA/B, and the UL16-binding proteins or CD155 and CD112, respectively. Further, expression of the inhibitory receptor NKG2A is known to impact NK effector responses through its interaction with HLA-E molecules (Bottino et al., 2005). In this sense, therapeutic interventions blocking this interaction represent promising tools to potentiate NK cell immune responses during different pathologies (André et al., 2018; Pereira et al., 2019). Importantly, HIV infection causes NK cell dysfunction, which is not completely restored by ART (Nabatanzi et al., 2019; Lichtfuss et al., 2012). NK cells play an important role in containing viral replication during early infection and shaping adaptive immune responses during chronic infection (Flórez-Álvarez et al., 2018). However, the role of NK cells in controlling the viral reservoir in PLWH on ART remains undefined. Mounting evidence supports the importance of NK cell function in shaping the HIV reservoir size, for example, through the induction of interferon (IFN-ɣ) and the expression of certain activating receptors (Marras et al., 2017). Moreover, proportions of NK cells inversely correlated with HIV-1 DNA reservoir levels in a clinical study designed to disrupt viral latency (Olesen et al., 2015). Importantly, during chronic infection, several strategies used by HIV to evade NK cell immune responses have been described, including the modulation of HLA class I molecules expression at the surface of infected cells (Jost and Altfeld, 2012). However, immune evasion mechanisms of HIV-infected reservoir cells to NK cell-mediated killing have not been identified.

Here, we show that CD32 expression on HIV-infected cells confers a reduced susceptibility to NK cell-mediated ADCC killing by a mechanism compatible with a reduced binding of HIV-specific antibodies required for this mechanism. Importantly, this immune-resistant mechanism is also observed in latently HIV-infected cells from ART-treated PLWH after viral reactivation, providing a plausible explanation for the maintenance of transcriptionally active HIV-infected cells in ART-treated PLWH.

## Results

### Susceptibility of HIV-reservoir cell subsets to NK immune response

While it is clear that ADCC, largely mediated by NK cells, is an important protective mechanism against HIV and simian immunodeficiency virus (SIV) infection (**Alpert et al., 2012**; **Pollara et al., 2011**; **Haynes et al., 2012**), the capacity of this immune mechanism to limit the infection of different T cell subsets composing the viral reservoir is currently unknown. Thus, we first assessed the intrinsic susceptibility of Naïve ($T_{NA}$), Stem Cell Memory ($T_{SCM}$), Central Memory ($T_{CM}$), Effector Memory ($T_{EM}$), $T_{CD20}{}^{dim}$, and $T_{CD32}{}^{dim}$ CD4$^+$ T cell subsets to ADCC response. Gp120-coated CD4$^+$ T cells from 15 ART-treated and virologically-suppressed PLWH (participants #8–22, **Supplementary file 1**) were subjected to a flow cytometry-based ADCC assay (**Gómez-Román et al., 2006**). In this assay, we used gp120-coated primary CD4$^+$T cells, and plasma from an HIV-infected person with a high titer of HIV-specific immunoglobulins, which allowed a comparative evaluation of the intrinsic susceptibility of the different subpopulations to NK-mediated killing. Gating of CD32 cells was performed as previously reported in our previous publications (**Grau-Expósito et al., 2017**), whereby possible contaminant cell conjugates with monocytes or B cells (defined as CD4$^+$ CD32$^{high}$ cells), were excluded. Coating efficiency with the recombinant protein gp120 of different cell subsets is shown in **Figure 1—figure supplement 1A**, and a representative gating strategy used for the identification of killed cells by ADCC is shown in **Figure 1—figure supplement 1B**. Results showed that each CD4$^+$ T cell subset had a different susceptibility to autologous NK cells, being $T_{EM}$ >$T_{CM}$ > $T_{CD20}{}^{dim}$>$T_{NA}$>$T_{SCM}$>$T_{CD32}{}^{dim}$ more prone to be killed (ANOVA Friedman test p=0.0001) (**Figure 1A**). Overall, most CD4$^+$T cell subsets were susceptible to ADCC, however $T_{CD32}{}^{dim}$ cells, and to a lesser extent $T_{SCM}$, showed the highest resistance to ADCC.

To address if the pattern observed for CD32 cells was exclusive of ART-treated PLWH, we included samples from Elite controllers (EC) and healthy donors (HD). First, we compared the expression of CD32 in CD4$^+$ T cells in the three cohorts (participants #1–22, **Supplementary file 1**), following a very stringent flow cytometry gating strategy (**Figure 1—figure supplement 1B**). In agreement with previous studies (**Darcis et al., 2020**; **Abdel-Mohsen et al., 2018**; **Holgado et al., 2018**; **García et al., 2018**), we found that a median of 1.21% of CD4$^+$ T cells expressed the CD32 receptor, and no statistically significant differences were detected between ART-suppressed, EC participants and HD (**Figure 1—figure supplement 1C**). In addition, and in concordance with the study of García et al., but differing from the report of Darcis et al. (**Darcis et al., 2020**; **García et al., 2018**), there was no correlation between the HIV-DNA or HIV-RNA levels and the frequency of CD32 expression on CD4$^+$ T cells (**Figure 1—figure supplement 1D–E**). We then performed the NK-ADCC assays. We observed that overall, NK cells from healthy donors were highly efficient at killing the total CD4$^+$ T cell population (**Figure 1B**) and the different CD4$^+$ T subsets (**Figure 1C**). EC represented a heterogeneous group of individuals, in which no significant differences were detected compared to HD as a total (**Figure 1B**) or by subset (**Figure 1C**). Of note, we observe a highly variable functionality of NK cells in the EC cohort compared with HD. We hypothesize that this variability in the EC cohort may be due to a higher diversity in the expression of NK receptors among individuals. More research is needed in this regard. Total CD4$^+$ T cells (**Figure 1B**) as well as all subsets from ART-treated PLWH were less susceptible to ADCC-mediated killing than HD (**Figure 1C**). The altered frequencies, phenotypes, and decreased functions of NK cell subsets reported during HIV infection (**Mikulak et al., 2017**), which are not fully restored by ART (**Lichtfuss et al., 2012**), could explain the different NK potency observed in these ART-suppressed PLWH. Notably, and regardless of that, NK cells from all cohorts showed a marked impaired capacity to kill $T_{CD32}{}^{dim}$ cells (median % of ADCC of 0.00, 34.84, and 45.01, for ART, EC, and HD, respectively) (**Figure 1C** and **Figure 1—figure supplement 1F–G**). We also investigated the potential relationship between the total cell reservoir size in vivo and ADCC activity. We observed a statistically significant inverse correlation between the percentage of ADCC activity against CD4$^+$ T cells and the total HIV-DNA reservoir size (**Figure 1D**) and, in particular, for the ADCC against $T_{CD32}{}^{dim}$ cells (**Figure 1E**).

HIV proteins are known to alter the expression of molecules on the infected cells, thereby impacting their recognition and likely the killing mediated by immune cells. Thus, we ought to confirm our results in a more physiological setting using ex vivo infected CD4$^+$ T cells. Isolated CD4$^+$ T cells from ART-suppressed PLWH were infected with HIV$_{BaL}$ or HIV$_{NL4.3}$, and after 5 days, were subjected to NK NC and ADCC assays (participants #23–29, 36, 44, 46–49, and 51, **Supplementary file 1**). A representative

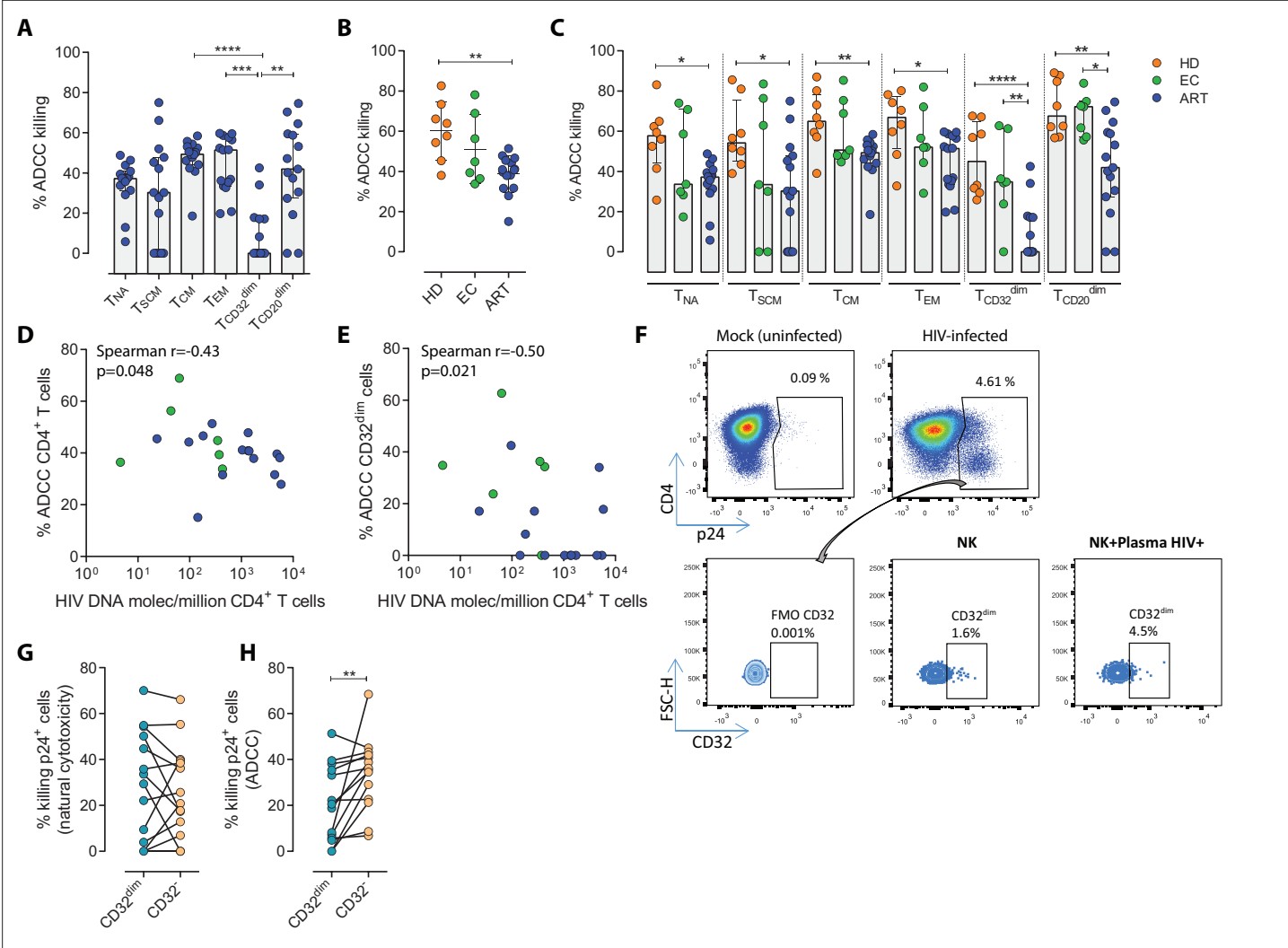

**Figure 1.** Susceptibility of CD4+ T cell subsets to the Natural Killer (NK) immune response. The susceptibility of different cell subpopulations that compose the HIV-reservoir to Natural Cytotoxicity (NC) and Antibody-Dependent Cell Cytotoxicity (ADCC) mediated by NK cells was measured by performing functional assays. (**A**) Percentage of gp120-coated cells killed by ADCC after being exposed to HIV-specific immunoglobulins (Igs) in the presence of NK cells. The intrinsic susceptibility to ADCC was measured in Naïve ($T_{NA}$), Stem Cell Memory ($T_{SCM}$), Central Memory ($T_{CM}$), Effector Memory ($T_{EM}$), $T_{CD32}^{dim}$, and $T_{CD20}^{dim}$ subsets. Statistical comparisons were performed using the ANOVA Friedman with Dunn's multiple comparison test. Median with interquartile range is shown. (**B**) Percentage of total gp120-coated CD4+ T cells from different cohorts of patients killed by ADCC. Healthy donors (HD), Elite Controllers (EC), and antiretroviral-treated (ART) PLWH. Statistical comparisons were performed using one-way ANOVA with Tukey's multiple comparison test. Median with interquartile range are shown. (**C**) Percentage of cell subsets killed by ADCC in cells from HD, EC, and ART. Statistical comparisons were performed using the one-way ANOVA with Tukey's multiple comparison test. Median with interquartile range is shown. (**D–E**) Spearman correlations between the size of the HIV-reservoir measured as total HIV-DNA in samples from ART-suppressed PLWH, and the potency of autologous NK cells to kill (**D**) total CD4+ T cells or (**E**) $T_{CD32}^{dim}$ cells by ADCC. (**F**) Representative flow cytometry gating strategy used to quantify HIV infection after ex vivo infection with BaL or NL4.3. Fluorescence minus one (FMO) control was used to determine CD32 expression. Cells were infected for 5 days and the frequency of expression of CD32 on HIV-infected cells was measured for each condition. (**G**) Percentage of killing by NC of ex vivo HIV-infected $T_{CD32}^{dim}$ and $T_{CD32}^{-}$ cells mediated by autologous NK cells from ART-treated PLWH (n=14). Killing was calculated by normalizing the proportion of each subset within the p24+ fraction in the co-culture condition to the basal condition. (**H**) Percentage of ADCC killing of ex vivo HIV-infected $T_{CD32}^{dim}$ and $T_{CD32}^{-}$ cells mediated by autologous NK cells from ART-treated PLWH (n=14). Killing was calculated by normalizing the proportion of each subset within the p24+ fraction in the co-culture condition with plasma to the co-culture without plasma. Statistical comparisons were performed using the Wilcoxon matched-pairs signed-rank test. *p<0.05; **p<0.01.

The online version of this article includes the following source data and figure supplement(s) for figure 1:

**Source data 1.** This file contains the source data used to generate **Figure 1**.

*Figure 1 continued on next page*

*Figure 1 continued*

**Figure supplement 1.** Gp120 cell coating efficiency, gating strategy for the NK-killing assays, and percentages of antibody-dependent cell cytotoxicity (ADCC) killing in elite controllers (EC), and healthy donors (HD).

**Figure supplement 2.** Percentage of CD32 expression on uninfected cells (p24⁻) and HIV-infected cells (p24⁺) with BaL (n=26) in (**A**) or with NL4.3 (n=11) in (**B**) after 5 days of infection.

flow gating strategy is shown in *Figure 1F*. After the ex vivo infection of CD4⁺ T cells, we observed higher expression of the CD32 molecule in comparison to uninfected cells (*Figure 1—figure supplement 2A–B*). Moreover, no significant differences between the killing of $T_{CD32}^{dim}$ and $T_{CD32}^{-}$ by NC in ART-suppressed PLWH were observed (*Figure 1G*). However, and in concordance with results in *Figure 1A and C*, $T_{CD32}^{dim}$ cells from ART-suppressed PLWH were significantly more resistant to ADCC in comparison to their negative cell counterparts (median % ADCC killing normalized to NC of 19.66 vs 35.85 for $T_{CD32}^{dim}$ and $T_{CD32}^{-}$, respectively) (*Figure 1H*). In addition, we observed that the capacity to kill infected $T_{CD32}^{dim}$ cells was directly related to the global capacity to kill all infected cells, indicating the essential role of NK potency, in addition to the intrinsic susceptibility of the target cells (*Figure 1—figure supplement 2C*). These results show that, regardless of differences between individuals on the overall killing capacity of their NK cells, different subpopulations of infected CD4⁺ T cells have distinct intrinsic susceptibility to ADCC responses, being the pool of CD4⁺ T cells expressing the CD32 molecule more resistant. Moreover, ART-suppressed individuals, most likely due to the existence of impaired NK cells, have a remarkable inability to kill this population of infected cells.

## Viral-reactivated cells expressing CD32 from ART-treated PLWH are resistant to NK cell-mediated cytotoxicity

Next, we examined if this NK-resistant profile might also affect the latent reservoir after viral reactivation. First, using samples from nine ART-suppressed PLWH (subjects #59–67, *Supplementary file 1*), we reactivated primary CD4⁺ T cells with different latency-reversing agents (LRA) and applied the Prime Flow RNA (FISH-flow) in situ hybridization (ISH) assay, which allows detecting at a single cell level cells expressing viral RNA, as described before (*Grau-Expósito et al., 2017*; *Grau-Expósito et al., 2019*). The gating strategy is shown in *Figure 2—figure supplement 1A*. We observed a higher increase of HIV-RNA⁺ cells in the $T_{CD32}^{dim}$ fraction compared to $T_{CD32}^{-}$ cells in all conditions and, within $T_{CD32}^{dim}$ cells, HIV-RNA⁺ cells were more frequent after Romidepsin treatment compared to untreated cells (*Figure 2A*). Upregulation of CD32 expression upon the treatment with several LRAs was evidenced in total CD4⁺ T cells (*Figure 2—figure supplement 1B*). Furthermore, we performed functional NK assays using samples from 22 additional ART-suppressed PLWH after reactivation of the natural HIV reservoir (participants #68–89, *Supplementary file 1*). First, we evaluated viral reactivation by intracellular p24 detection in CD4⁺ T cells (*Figure 2—figure supplement 1C*) and successfully observed viral reactivation of dormant HIV in 17 of these samples (*Figure 2B*). An example of p24 detection using this functional assay is shown in *Figure 2—figure supplement 1D*. Also in line with our previous results, we found a higher frequency of viral reactivated cells (p24⁺ cells) within the $T_{CD32}^{dim}$ fraction (*Figure 2C*). After the NK killing assays, viral-reactivated cells, in general, were susceptible to ADCC (*Figure 2D*, ART+LRA+NK+plasma condition) with a median of 30% increase in the ADCC activity compared with the NC activity (*Figure 2E*). Of note, we used autologous plasma from the same virologically-suppressed PLWH (mean time of undetectable viremia of 60 months [range 25–118]). This is noteworthy since ART treatment may decrease the number of antibodies mediating ADCC (*Madhavi et al., 2015*). We next evaluated the population of viral-reactivated cells expressing CD32 in these PLWH. We observed an increase in the expression of CD32 after latency disruption, which constituted a significant fraction of the total pool of viral reactivated cells (*Figure 2F*). Remarkably, this population was refractory to ADCC and even increased after the ADCC assays (condition with plasma) (*Figure 2G*). Thus, concordantly with previous results, $T_{CD32}^{dim}$ cells showed a higher pattern of ADCC resistance compared to the total HIV-reactivated cells (*Figure 2H*). Altogether, our results show that the latent HIV reservoir expresses CD32 upon viral reactivation with LRAs, and the resulting $T_{CD32}^{dim}$-infected cells are less sensitive to NK-mediated ADCC killing than the whole infected population. Importantly, antibodies endowed with ADCC-triggering capacity are still present in some ART-suppressed PLWH, yet the $T_{CD32}^{dim}$ population might escape from this immune mechanism.

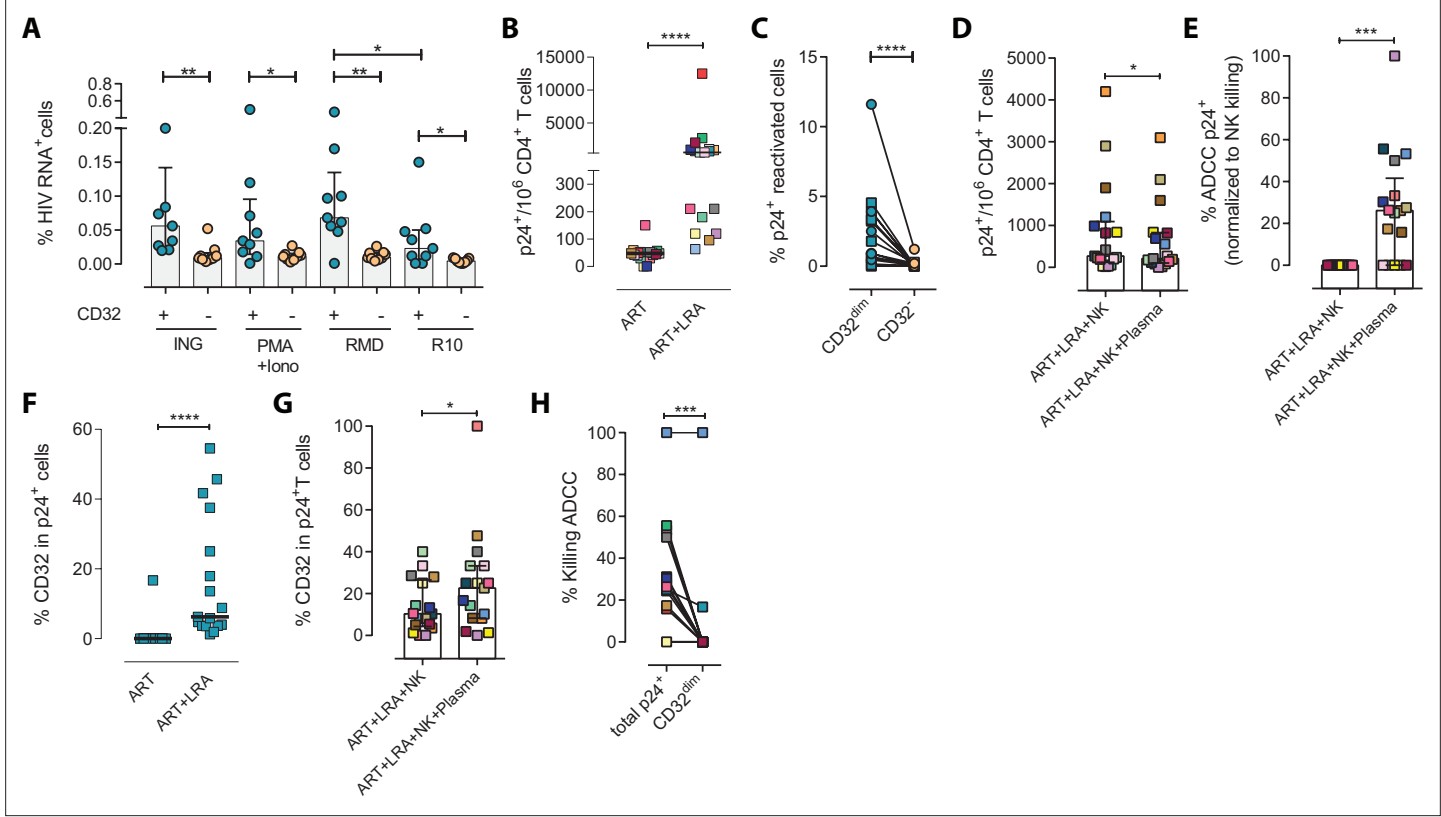

**Figure 2.** Expression of HIV in $T_{CD32}^{dim}$ reservoir cells after latency disruption and susceptibility to natural killer (NK) immune responses. Data from the direct ex vivo reactivation of the natural HIV reservoir in ART-suppressed PLWH. (**A**) Percentage of HIV-RNA expressing cells, measured by the RNA FISH-flow assay, within the $T_{CD32}^{dim}$ and $T_{CD32}^{-}$ subsets after viral reactivation with Ingenol, PMA/ionomycin, or romidepsin (n=9). Statistical comparisons were performed using the Wilcoxon matched-pairs signed-rank test (comparison between CD32$^{dim}$ and CD32$^{-}$ within each drug condition), and the ANOVA Friedman with Dunn's multiple comparison test (comparison between different drug conditions of the cell subset). Median with interquartile range is represented. (**B**) Percentage of p24$^{+}$ cells after 18 hr viral reactivation with PMA/ionomycin (n=17). Each participant is represented by a different color. Statistical comparisons were performed using the Wilcoxon matched-pairs signed-rank test. (**C**) Frequency of viral reactivation within the total pool of $T_{CD32}^{dim}$ and $T_{CD32}^{-}$ cells (n=17). Statistical comparisons were performed using the Wilcoxon matched-pairs signed-rank test. (**D**) NK killing assays against viral reactivated cells. Number of p24$^{+}$ cells per million CD4$^{+}$ T cells after the addition of NK cells only or together with the autologous plasma is shown (n=17). Statistical comparisons were performed using the Wilcoxon matched-pairs signed-rank test. (**E**) Percentage of ADCC in p24$^{+}$ cells normalized to the NC control (ART + LRA + NK). Statistical comparisons were performed using one-sample t-test. (**F**) Percentage of CD32 expression within the total p24$^{+}$ pool before and after HIV reactivation (n=17). Statistical comparisons were performed using the Wilcoxon matched-pairs signed-rank test. (**G**) Percentage of $T_{CD32}^{dim}$ within p24$^{+}$ cells after HIV reactivation and functional NK-mediated assays (n=17). Statistical comparisons were performed using the Wilcoxon matched-pairs signed-rank test. (**H**) Percentage of NK-mediated killing by ADCC of the reactivated $T_{CD32}^{dim}$ or total p24$^{+}$ cells (n=17). ADCC was calculated as the reduction of p24$^{+}$ cells after the co-culture with NK and plasma and normalized to the condition with NK cells alone. Statistical comparisons were performed using the Wilcoxon matched-pairs signed-rank test. *p<0.05; **p<0.01; ***p<0.001.

The online version of this article includes the following source data and figure supplement(s) for figure 2:

**Source data 1.** This file contains the source data used to generate *Figure 2*.

**Figure supplement 1.** Expression of HIV-RNA and viral protein p24 in $T_{CD32}^{dim}$ reservoir cells after viral reactivation.

## Infected $T_{CD32}^{dim}$ cells expressing activating or inhibitory NK ligands are more refractory to NK-mediated killing

To ascertain if differences in receptor-ligand interactions could be responsible for the impaired capacity of the NK cells to kill the $T_{CD32}^{dim}$ subset, we studied the expression of MICA/B, ULBP-1, CD155, and HLA-E on the surface of HIV-infected CD4$^{+}$ T cells after ex vivo infection. Of note, the impact of HIV infection on the expression of many of these ligands is not fully understood and seems to depend on the viral strain and the stage of the viral infection (*Tremblay-McLean et al., 2017*; *Apps et al., 2016*). Expression of the ligands was assessed by flow cytometry, and the gating strategy used for these analyses is shown in *Figure 3—figure supplement 1A*. Overall, we observed that, despite HIV

induced higher expression of MICA/B, ULBP-1, and CD155 in infected cells compared to uninfected cells, only a small proportion of infected cells expressed these activating ligands (*Figure 3A and B*). In contrast, the ligand HLA-E was found to be expressed in a significantly higher proportion of infected cells (median of 12.0% in HIV-infected cells vs 9.4% in uninfected cells, p=0.002) (*Figure 3C and D*). By analyzing CD4$^+$ T-infected cells based on their expression of CD32, the same trend was observed for the receptors MICA/B, ULBP1, or CD155 (median 0.6%, 1.28%, 0.37% for infected $T_{CD32}{}^{dim}$ vs 0.26%, 0.09%, 0.26% for the infected $T_{CD32}{}^-$ fraction) (*Figure 3E and F*). However, HLA-E expression was 2-fold significantly higher in HIV-infected $T_{CD32}{}^{dim}$ compared to infected $T_{CD32}{}^-$ cells (median of 22.2% and 11.0%, respectively, p<0.0001), and differences in MFI were also detected (*Figure 3G and H*). Of note, in the absence of any viral infection, $T_{CD32}{}^{dim}$ cells showed intrinsically higher HLA-E expression (*Figure 3—figure supplement 1B*).

Next, we ought to determine the expression of activating and inhibitory ligands on the fraction of CD4$^+$ T cells refractory to ADCC after ex vivo infection. We performed both NC and ADCC functional assays (in cells from participants #36, 40, 44, 45, 47–49, and 51, *Supplementary file 1*). We observed statistically significant differences (ANOVA Friedman test) for the infected population (p24$^+$) expressing the ligands ULBP-1 (p=0.004), MICA/B (p=0.030), and CD155 (p=0.017) after NC and ADCC assays (*Figure 3I–K*); however, only the cells expressing the ligand ULBP-1 in the NC and ADCC condition were different compared to the basal (*Figure 3I*). Of note, as previously reported (*Pereira et al., 2019*; *Ward et al., 2004*), cells expressing the molecule HLA-E were particularly resistant to NK-mediated killing (*Figure 3L*). However, infected $T_{CD32}{}^{dim}$ cells expressing any of the activating NK ligands were more refractory to both NK-mediated immune responses (*Figure 3M–P*). Overall, we observed that upon HIV infection, target cells expressing NK-activating ligands showed some susceptibility to NK cells. However, infected cells expressing HLA-E and, particularly, CD32$^{dim}$ cells expressing this molecule and/or the activating ligands MICA/B, ULBP-1, and CD155, were more resistant to NK-mediated killing.

## Suboptimal binding of HIV-specific immunoglobulins to CD32 in HIV-infected cells inefficiently triggers NK cell degranulation

Different hypotheses might help to explain why $T_{CD32}{}^{dim}$ cells are more resistant to ADCC. CD32a is a low-affinity receptor for the constant fraction of immunoglobulin G (FcγR-IIa) (*Alevy et al., 1992*; *Veri et al., 2007*; *Anania et al., 2019*), and the expression of this molecule on HIV-infected cells would provide them with a bivalent capacity to interact with immunoglobulins present in plasma, both through their constant fraction (Fc portion) or the variable fragment (antigen-specific). In addition, pentraxins, conserved innate immune molecules involved in infectious processes and inflammation, can bind to CD32 and compete with IgGs (*Lu et al., 2008*). Thus, the engagement of such molecules or immune complexes (IC) to the CD32 Fc receptor might offer a selective advantage by protecting infected cells from HIV-specific ADCC-inducing antibodies. Accordingly, we tested if HIV-specific immunoglobulins (Igs) were able to efficiently bind to CD4$^+$ T CD32-expressing cells previously coated with gp120. After incubation with plasma containing HIV-unspecific immunocomplexes (ICs), we found a decrease in the total number of HIV-specific molecules (A32 mAb) able to bind to $T_{CD32}{}^{dim}$ cells (*Figure 4A–B*), but such effect was not observed in the $T_{CD32}{}^-$ population (*Figure 4C*). A reduction in the total number of cells being recognized by the A32 mAb was not observed (*Figure 4—figure supplement 1A-B*). This suggests that components present in plasma might bind to the FcγR receptor CD32 representing a steric hindrance and precluding further binding of HIV-specific IgGs. Alternatively, this may indicate competition between CD32 and gp120 for the binding of the same IgG.

We also studied the capacity of ex vivo infected CD4$^+$ T cells expressing CD32 to form ADCC-induced conjugates with NK cells. We observed a significantly higher frequency of $T_{CD32}{}^{dim}$ – NK doublets compared to $T_{CD32}{}^-$ cells (*Figure 4D*). This could be explained by the inefficient elimination of $T_{CD32}{}^{dim}$ cells by ADCC (as shown in *Figure 1H*), more likely as a consequence of suboptimal antibody-induced immune synapse. This phenomenon has been previously described in HIV-infected macrophages where increased effector-target cell contact time was associated with relative resistance to cytotoxic T lymphocytes (*Clayton et al., 2018*). Indeed, when we tested the capacity of FACS-sorted and gp120-coated $T_{CD32}{}^{dim}$ cells to activate NK cells, we observed that these cells triggered a less potent NK degranulation than their negative counterparts (*Figure 4E*), whileIFN-γ was equally induced in both cases (*Figure 4—figure supplement 1C*). Overall, our results suggest that potential

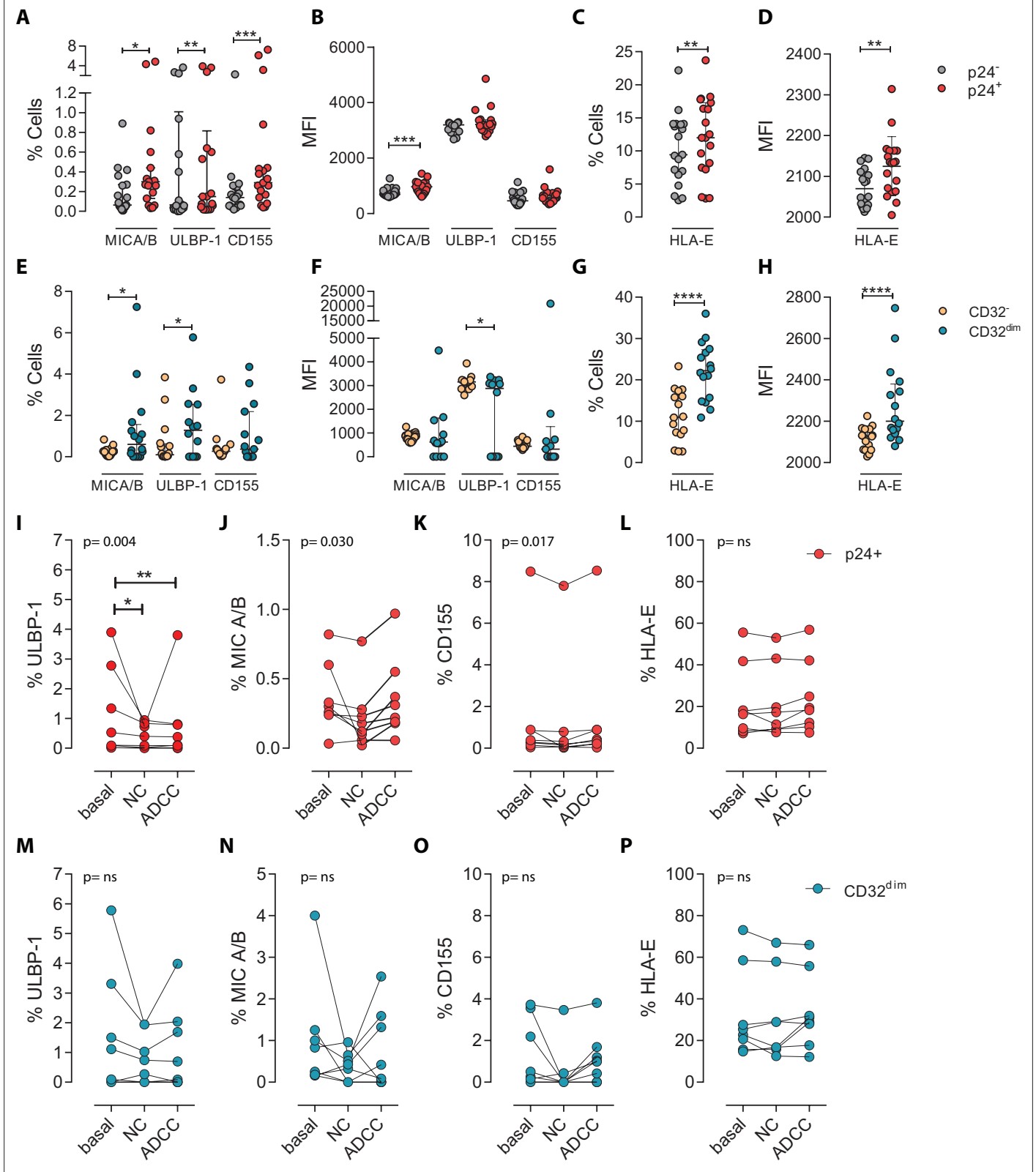

**Figure 3.** Expression of NK-ligands in cells resistant to NK-mediated killing. HIV-infected cells from healthy donors were subjected to NK-killing assays and the percentage of expression of different NK-ligands was measured by flow cytometry in different fractions (CD32⁻ and CD32ᵈⁱᵐ) of infected (p24⁺) or uninfected (p24⁻) cells. (**A–H**) Expression of NK-ligands before performing the killing assays (n=19). (**A**) Percentage of CD4⁺ T cells expressing MIC A/B, ULBP-1, and CD155. (**B**) Mean Fluorescence Intensity (MFI) values for the expression of MIC A/B, ULBP-1, and CD155 on CD4⁺ T cells. (**C**)

*Figure 3 continued on next page*

*Figure 3 continued*

Percentage of CD4$^+$ T cells expressing the MHC molecule HLA-E. (**D**) MFI values for HLA-E expression on CD4$^+$ T cells. All graphs show median with interquartile range and the statistical comparisons were performed using the Wilcoxon matched-pairs signed-rank test (comparison between p24$^+$ and p24$^-$). (**E–H**) Same analyses as A–D but showing HIV-infected CD32$^{dim}$ and CD32$^-$ cells. (**I–P**) Expression of NK-ligands on HIV-infected cells not killed by the different NK-killing mechanisms. Natural Cytotoxicity (NC) and Antibody-Dependent Cell Cytotoxicity (ADCC). (**I–L**) Expression of NK-ligands on total infected CD4$^+$ T cells before and after NK killing. (**M–P**) Expression of NK-ligands on infected T$_{CD32}$$^{dim}$ cells before and after NK killing. All I-P graphs show median with interquartile range. p values shown in the graphs represent ANOVA Friedman test, and asterisks denote the multiple comparison Dunn's test.*p<0.05; **p<0.01; ***p<0.001; ****p<0.0001.

The online version of this article includes the following source data and figure supplement(s) for figure 3:

**Source data 1.** This file contains the source data used to generate *Figure 3*.

**Figure supplement 1.** NK-ligands on HIV-infected cells.

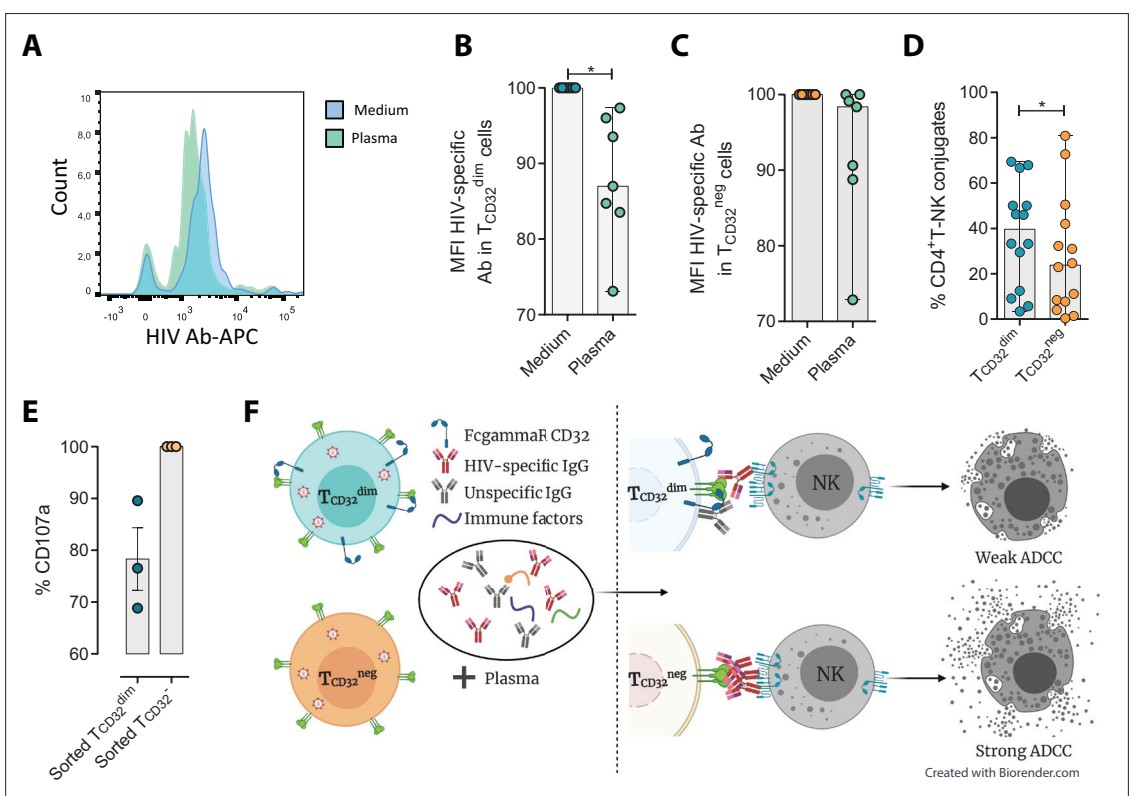

**Figure 4.** Reduced binding of HIV-specific antibodies to T$_{CD32}$$^{dim}$ cells and the effect on natural killer (NK) degranulation. The binding capability of the HIVgp120-specific IgG A32, an antibody (Ab) labeled with allophycocyanin (APC), to gp120-coated T$_{CD32}$$^{dim}$ and T$_{CD32}$$^-$ cells from healthy donors, before and after incubation with plasma containing non-HIV specific IgGs, was measured by flow cytometry (n=7). Percentage of the Mean Fluorescence Intensity (MFI) signal, normalized to the medium, for A32$^+$ cells after plasma addition is shown in (**A**) Representative histogram of A32$^+$ T$_{CD32}$$^{dim}$ cells, (**B**) T$_{CD32}$$^{dim}$ and (**C**) T$_{CD32}$$^-$ cells. (**D**) Percentage of cell conjugates between ex vivo HIV-infected T$_{CD32}$$^{dim}$ or T$_{CD32}$$^-$ and NK cells after performing antibody-dependent cell cytotoxicity (ADCC) assays (n=14). (**E**) Percentage of NK degranulation (CD107a marker) in cell conjugates with sorted T$_{CD32}$$^{dim}$ or T$_{CD32}$$^-$-coated with gp120 HIV protein and incubated with plasma HIV$^+$, after a 4 hr antibody-dependent cell cytotoxicity (ADCC) activation assay (n=3). Values are normalized to the CD32 population (**F**) Schematic illustration of the impaired ADCC response against T$_{CD32}$$^{dim}$ cells. Graphs show median with range and statistical comparisons were performed using Wilcoxon matched-pairs signed-rank test. *p<0.05.

The online version of this article includes the following source data and figure supplement(s) for figure 4:

**Source data 1.** This file contains the source data used to generate *Figure 4*.

**Figure supplement 1.** Binding of A32 to T cells and percentage of IFN-γ in cell conjugates.

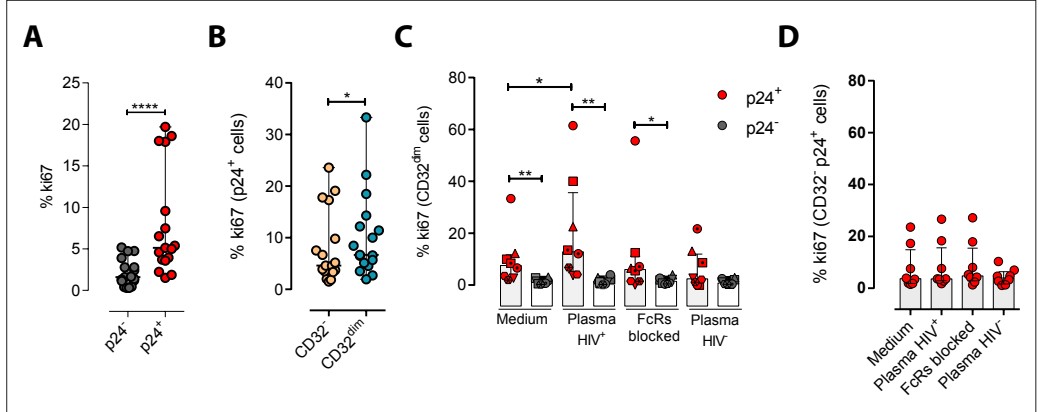

**Figure 5.** Immune complexes engagement with CD32 induces proliferation in HIV-i-nfected CD32$^{dim}$ cells. Cells from healthy donors were infected with the viral strain HIV$_{BaL}$ and 5 days post-infection Ki67 expression was measured by flow cytometry. (**A**) Expression of the proliferation marker Ki67 in uninfected or ex vivo HIV-infected CD4$^+$ T cells. Median and ranges are shown (n=17). Statistical comparisons consisted of the Wilcoxon matched-pairs signed-rank test. (**B**) Percentage of Ki67$^+$ cells in HIV-infected T$_{CD32}$$^{dim}$ and T$_{CD32}$$^-$ subsets are shown (n=17). Median and ranges are shown. Statistical comparisons consisted of the Wilcoxon matched-pairs signed-rank test. (**C**) Percentage of Ki67 expression on T$_{CD32}$$^{dim}$ cells after immune complexes engagement (Plasma HIV$^+$). Medium alone, FcRs blockers and plasma from an HIV-negative individual were included as controls. Median with interquartile range is shown (n=8). Statistical comparisons were performed using the Wilcoxon matched-pairs signed-rank test (comparison between p24$^+$ and p24$^-$ within each condition), and the ANOVA Friedman with Dunn's multiple comparison test (comparison between different conditions of the p24$^+$ or p24$^-$ cells). (**D**) Percentage of Ki67 expression on HIV-infected T$_{CD32}$$^-$ cells incubated under the same experimental conditions as shown in (**C**). Median with interquartile range is represented (n=8). Statistical comparisons consisted of the Wilcoxon matched-pairs signed-rank test.*p<0.05; **p<0.01; ***p<0.001; ****p<0.0001.

The online version of this article includes the following source data for figure 5:

**Source data 1.** This file contains the source data used to generate **Figure 5**.

immune factors present in the plasma that are ligands of CD32 might promote a steric interference that precludes the subsequent binding of HIV-specific antibodies, leading to poor NK activation and therefore resistance to ADCC (as illustrated in **Figure 4F**).

## HIV-infected cells expressing CD32 show higher proliferation potential after immune complexes engagement

We next explored the possibility that the engagement of IC to the CD32 molecule might induce cell proliferation, and therefore, contribute to their persistence. To study that, we performed ex vivo infection experiments of primary CD4$^+$ T cells and measured cell proliferation after IC engagement by flow cytometry. First, we observed that HIV-infected cells, and in particular infected cells expressing CD32$^{dim}$, had higher proliferative potential measured by the expression of Ki67 (median of 6.67% vs 4.62% for T$_{CD32}$$^{dim}$ and T$_{CD32}$$^-$, respectively) (**Figure 5A and B**). Importantly, we observed that the addition of plasma from an HIV-infected patient (containing a high titer of Igs), induced significant proliferation of infected T$_{CD32}$$^{dim}$ cells (median of 7.52% vs 12.80% for basal condition and plasma HIV$^+$, respectively), which was not significant when Fc receptor blockers were added to the culture (**Figure 5C**). Of note, no effect was observed after the addition of plasma from an HIV-negative donor (**Figure 5C**). In contrast, cell proliferation was unchanged in the infected T$_{CD32}$$^-$ population (**Figure 5D**). Overall, these results suggest that IC present in the plasma of PLWH could contribute to the perpetuation of the T$_{CD32}$$^{dim}$ HIV-infected subset by inducing cell proliferation, besides protecting them from ADCC.

## Effect of IL-15 and IFN-α at enhancing the killing of HIV-infected T$_{CD32}$$^{dim}$ cells

Given the unique properties of T$_{CD32}$$^{dim}$ cells, we further explored several strategies to potentiate its elimination by NK cells. We tried to directly reinvigorate NK cells from HIV-infected PLWH (#52–58,

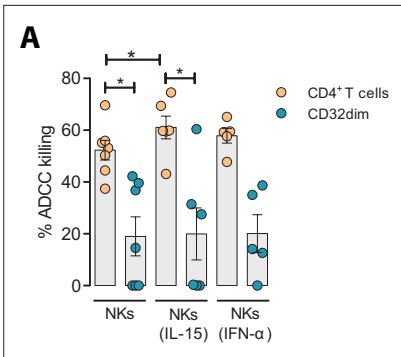

**Figure 6.** Effect of IL-15 and IFN-α in the natural killer (NK) function. CD4+ T cells from ART-suppressed participants coated with a gp120 recombinant protein were subjected to antibody-dependent cell cytotoxicity (ADCC) assays in the presence of cytokines. The graph shows the percentage of ADCC killing of HIVgp120-coated CD4+ T cells by autologous NK cells after treatment with IL-15 or IFN-α. Statistical comparisons were performed using the Wilcoxon matched-pairs signed-rank test (comparison between CD4+ T and CD32dim cells within each condition), and the ANOVA Friedman with Dunn's multiple comparison test (comparison between different conditions of the CD4+ T or CD32dim cells). Mean with SEM is represented. *p<0.05.

The online version of this article includes the following source data for figure 6:

**Source data 1.** This file contains the source data used to generate *Figure 6*.

*Supplementary file 1*) culturing them with the cytokines IL-15 or IFN-α. As expected and previously reported (*Garrido et al., 2018*), we observed an enhanced performance of NK cells against the total gp120-coated CD4+ T cell population when treated with IL-15, and to a lesser extent with IFN-α. However, neither IL-15 nor IFN-α was able to enhance the ADCC response against T $_{CD32}^{dim}$-infected cells (*Figure 6A*).

## Discussion

The presence of cellular and anatomical viral reservoirs, not susceptible to ART or antiviral immune responses, is the main barrier to cure the HIV infection. Thus, elucidating how these reservoirs are maintained for prolonged periods of time represents an important step towards the cure of HIV. Remarkably, we postulate a pivotal role of ADCC-NK in shaping the HIV reservoir during ART, being a novel route to avoid NK cell effector immunity by HIV-reservoir cells. We found that the expression of the molecule CD32 on productively HIV-infected cells is associated with a reduced susceptibility to ADCC activity by NK cells. Furthermore, when infected ex vivo, these cells express higher levels of the molecule HLA-E, which also limits NK-mediated killing. Last, upon interaction with immune complexes, their capacity to bind to HIV-specific antibodies

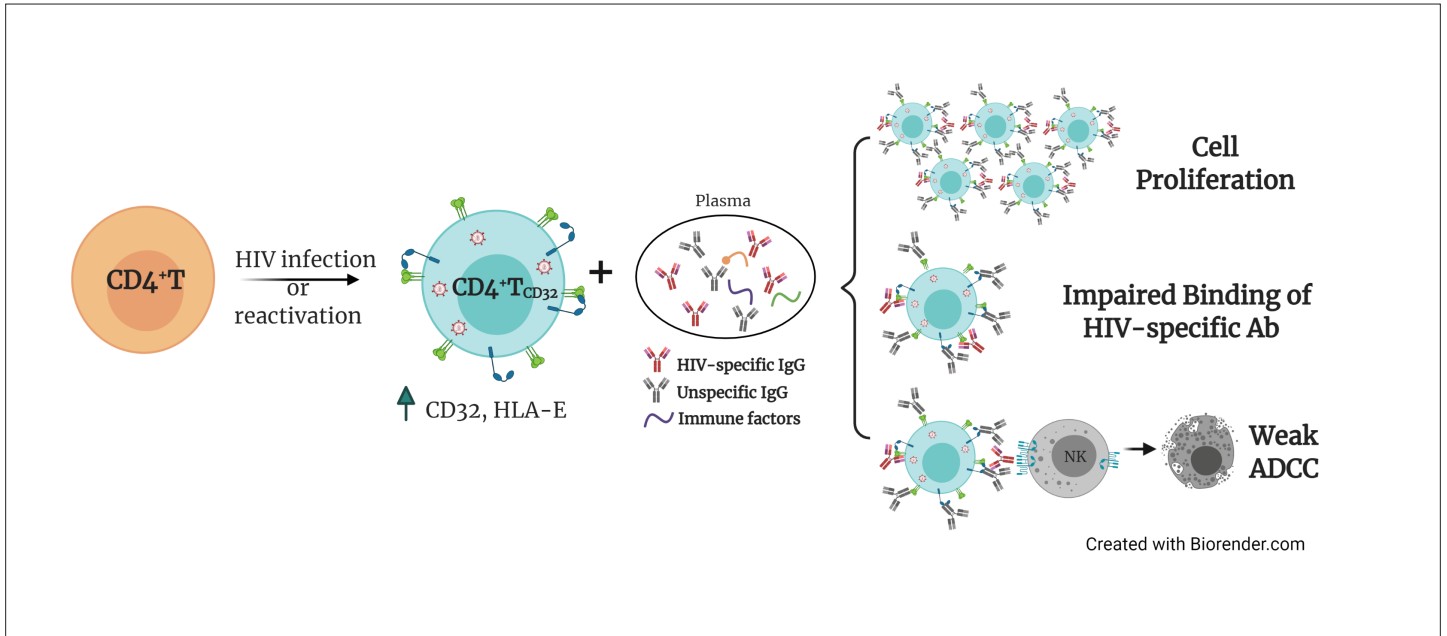

**Figure 7.** Proposed model for the persistence of CD4+ T CD32dim cells. A proportion of CD4+T cells will express CD32 molecules upon HIV infection or viral reactivation (spontaneous reactivation or after the use of latency reversal agents). CD4+T CD32dim cells express higher levels of HLA-E, a molecule that inhibits NK cells expressing the receptor NKG2A, and are more susceptible to immune factors present in the plasma, which will (i) induce cell proliferation, (ii) block the recognition of HIV-specific antibodies, and (iii) induce a weak antibody-dependent cell cytotoxicity (ADCC) response.

decreases, while gaining the potential to proliferate. Considering that a significant proportion of HIV-latently infected cells expressed CD32 upon viral reactivation, as shown here and reported before (**Grau-Expósito et al., 2019**), these cells will indeed benefit from these described NK immune evasion mechanisms. Altogether, these factors may greatly contribute to perpetuating the persistence of the cell reservoir upon viral expression, either after spontaneous viral reactivation or after the use of latency reversal agents. A schematic representation of the proposed model for the persistence of CD4+T CD32dim cells is shown in **Figure 7**.

One of the main obstacles in the HIV cure field has been the lack of reliable markers to uniquely identify persistently infected cells. Among proposed molecules, we find immune checkpoint inhibitors such as PD-1 (**Fromentin et al., 2016**), the B cell surface marker CD20 (**Serra-Peinado et al., 2019**), CD30 (**Hogan et al., 2018**), or, more recently, a combination of several receptors (**Neidleman et al., 2020**). However, the Fcγ receptor CD32 is perhaps one of the most promising HIV reservoir markers, since it is expressed during latent (**Descours et al., 2017**; **Darcis et al., 2020**) and transcriptionally-active infection (**Abdel-Mohsen et al., 2018**; **Vásquez et al., 2019**; **Noto et al., 2018**; **Huot et al., 2021**); it has been localized in main tissue reservoirs such as the cervical tissue (**Cantero-Pérez et al., 2019**), lymph nodes (**Abdel-Mohsen et al., 2018**; **Noto et al., 2018**) and the gastrointestinal tract (**Vásquez et al., 2019**) and, in some cases, it has been associated with a very prominent enrichment for HIV DNA (**Cantero-Pérez et al., 2019**; **Descours et al., 2017**; **Darcis et al., 2020**). This molecule is typically expressed on myeloid cells or platelets, in which its function has been extensively studied (**Anania et al., 2019**). While CD32 expressed on CD4+ T cells is fully functional (**Holgado et al., 2018**; **Engelhardt et al., 1995**), many questions remain unknown regarding the expression dynamics and function, in particular during HIV pathogenesis. In a recent study with SIV-infected non-human primates, CD32+CD4+ T cells were strongly increased in LNs, spleen, and intestine during SIV mac infection, were enriched in markers often expressed on HIV-infected cells, and contained higher levels of actively transcribed SIV RNA (**Huot et al., 2021**). Consistent with previous reports (**Abdel-Mohsen et al., 2018**; **García et al., 2018**), we detected dim levels of CD32 on CD4+ T cells which increased upon ex vivo HIV infection or reactivation. This result is in line with a publication showing the ability of cells expressing CD32 to reactivate latent HIV (**Darcis et al., 2020**). Moreover, tissue-resident CD4+ T cells with expression of CD32 have been reported in cervical samples in the absence of infection. Importantly, this fraction of cells was intrinsically enriched for the expression of molecules related to HIV susceptibility and long-term maintenance (**Cantero-Pérez et al., 2019**). In this sense, cell proliferation is one of the most important mechanisms of cell reservoir maintenance in long-term ART-suppressed PLWH (**Chomont et al., 2009**; **Gantner et al., 2020**; **Simonetti et al., 2016**). Importantly, we show that CD32-expressing cells had a higher proliferative potential in response to IC. Inherent proliferative capacity of this subset has recently been reported in a study demonstrating CD4+ T cell activation upon CD32 ligation with antibodies or aggregated IgG (**Holgado et al., 2018**). Thus, understanding the mechanisms by which HIV-infected cells expressing CD32 are maintained in the human body could significantly advance the search for an HIV cure.

NK cells are key players in the defense against many pathogens, including HIV, being not only one of the first lines of protection but also essential modulators of the adaptive immune responses. Therefore, the acquisition of resistance mechanisms to any of the NK effector functions may contribute to pathogen survival and, in the case of HIV, favor conditions for viral persistence. In this sense, it has been reported that NK cell immune pressure leads to viral sequence evolution (**Alter et al., 2011**), and HLA-mediated immune resistance mechanisms have been previously identified in productively HIV-infected cells (**Cohen et al., 1999**; **Bonaparte and Barker, 2004**). Moreover, resistance of reservoir cells to HIV-specific cytotoxic T cells has also been reported (**Ren et al., 2020**; **Huang et al., 2018**), and Nef has been identified as a key protein for the persistence of genetically intact HIV proviruses in effector memory CD4+ T cells (**Duette et al., 2022**), suggesting that viral persistence might be facilitated not only by cell proliferation mechanisms in the absence of viral antigen expression, but also by avoiding immune-mediated killing. NK cells recognize IgG–viral protein complexes, namely immune complexes, on infected cells via FcγRs to mediate ADCC, which is a potent mechanism to eliminate virally infected cells (**Forthal and Finzi, 2018**). However, HIV has developed several strategies to evade this immune response. For instance, Vpu reduces the presence of viral antigens susceptible to recognition by antibodies on the surface of infected cells (**Arias et al., 2014**). This accessory protein is responsible for the decreased expression of tetherin, a cellular host restriction factor that retains

HIV virions on the cell surface, and therefore diminishes ADCC responses (*Arias et al., 2014*). Moreover, Vpu and Nef downregulate CD4 expression on infected cells, preventing its interaction with Env trimers, which subsequently impedes the binding of ADCC-antibodies (*Veillette et al., 2014*). Overall, a proper antibody-induced immune synapse with NK cells is required to elicit a potent ADCC immune response, which depends on many factors, such as spatial configuration, valence of the antibody-epitope binding, antibody conformation, and the resulting size of the immune complex (*Murin, 2020*). In our study, we show that the interaction of $T_{CD32}^{dim}$ cells with IC present in plasma leads to suboptimal binding of HIV-specific antibodies, limiting ADCC. These results suggest that potential Igs, and likely other immune mediators such as pentraxins (*Lu et al., 2008*), might contribute to maintaining HIV-reservoir cells through the interaction with the CD32 receptor, conferring a diminished susceptibility to ADCC activity, which intrinsically requires the presence of immune complexes. This is not the first non-desirable effect of Igs identified in the context of an infectious disease; an antibody-dependent enhancement of infection has been extensively reported for Dengue infection (*Martina et al., 2009*). The intrinsic nature of antibodies, with an antigen-binding fragment (Fab) and a crystallizable fragment (Fc) as separate functional domains, adds complexity to the multiple functions and interactions that ultimately contribute to modulate immunity (*Lu et al., 2018*). In fact, a major line of research in the field currently focuses on the use of broadly neutralizing antibodies (bNAbs) as a therapeutic or prophylactic treatment for HIV infection, relying on the efficient blocking of the virus and the triggering of potent immune effector responses, for example, through FcγR interactions (*Hessell et al., 2007*; *Bournazos et al., 2014*). The potentiation of the NK activity with these new approximations may help to overcome the intrinsic resistance of infected cells expressing the CD32 receptor.

Furthermore, as previously reported by others, HIV infection maintains or even increases the expression of the non-classical HLA-E molecule, which may contribute to the inhibition of the NK cytotoxic response (*Ward et al., 2004*; *Martini et al., 2005*). Interestingly, we found enrichment of HLA-E expression in the pool of HIV-infected cells expressing CD32, which could tilt the balance towards NK inhibition, therefore, conferring an additional advantage to survive. The effect of HLA-E suppressing immune responses has been shown in several settings, including in the context of senescence or tumors, where blocking its interaction with the inhibitory NK receptor NKG2A reinvigorated effector functions (*André et al., 2018*; *Pereira et al., 2019*). In addition, HLA-E has been shown to impair ADCC against HIV-expressing cells, while impeding its interaction with NK cells improved the elimination of target cells (*Ward et al., 2004*).

In recent years, major efforts have focused on the identification of compounds to reactivate persistent HIV from its dormant state, with the ultimate goal to eliminate viral infection. However, it has become clear that the stimulation of the immune system is also a mandatory step for the elimination of persistent HIV (*Ward et al., 2021*). Our results show that the magnitude of the killing depended on the potency of the NK cells, where NK cells from ART-suppressed PLWH with higher total HIV reservoir size were functionally impaired. These results are in agreement with previous reports showing that HIV chronic infection has a deleterious effect on NK cell function, which is not completely restored despite ART (*Nabatanzi et al., 2019*; *Lichtfuss et al., 2012*). To reinvigorate NK function in these subjects, we treated NK cells with the cytokines IL-15 or IFN-α. Despite a clear immune effect on the total infected population, we did not increase the killing of $T_{CD32}^{dim}$ cells. A plausible explanation would be the upregulation of NKG2A on NK cells mediated by these cytokines (*Merino et al., 2019*; *Mori et al., 1998*), which would further limit the killing of cells already expressing high levels of HLA-E, such as $T_{CD32}^{dim}$ cells.

A potential limitation of our study is the decline of HIV-specific antibody levels in ART-treated PLWH (*Madhavi et al., 2015*; *Jensen et al., 2015*). Despite this decline, functional ADCC activity mediated by NK cells remains detectable in long-term virally-suppressed individuals (*Madhavi et al., 2015*). Of relevance, we show here the existence of functional ADCC activity using plasma samples from virologically-suppressed PLWH. These antibodies induced significant NK-mediated elimination of the total pool of reactivated latently infected cells in many samples, although the subpopulation expressing CD32 was more resistant. Thus, our results suggest that, after viral reactivation, levels of HIV antigen expression susceptible to recognition by immune cells such as NK may be induced. Also, that ADCC could play a significant role in HIV persistence, highlighting the need for new strategies directed to impact resistant reservoir cells such as $T_{CD32}^{dim}$ cells. However, the limited number of these cells observed in vivo in ART-treated PLWH restrains their study. In this sense, the few viral-reactivated

cells detected after LRA treatment from the natural latent reservoir impede an extensive assessment of infected cells expressing CD32. Further, it is unclear which is the dynamic of CD32 expression on individual cells after viral reactivation and if changes on its expression would affect ADCC activity. Last, it is possible that infected $T_{CD32}^{dim}$ cells are susceptible to cytotoxic $CD8^+$ T lymphocytes. More research on these aspects are warranted. However, based on our results and regardless of the abundance of susceptible $CD32^{neg}$ cells that could contain latent HIV, we may speculate that this sort of immune resilient infected cells in vivo could certainly contribute to the source of viral rebound when ART is interrupted, or they could initiate a low level of viral replication in tissues where drugs are not able to fully penetrate. Moreover, their proliferative capacity in response to IC situates these cells as candidates for cells supporting cellular proliferation, one of the main mechanisms that perpetuate HIV reservoirs in vivo (*Sengupta and Siliciano, 2018*; *Halvas et al., 2020*).

# Materials and methods

## Key resources table

| Reagent type (species) or resource | Designation | Source or reference | Identifiers | Additional information |
|---|---|---|---|---|
| antibody | Human monoclonal A32 antibody | AIDS Research and Reference Program | Cat#11,438 | (1/200) |
| antibody | Mouse monoclonal anti-human CD32-PE-Cy7 (FUN-2) | Biolegend | 303,214 | (1/50) |
| antibody | Mouse monoclonal anti-human CD32-FITC (FUN-2) | Biolegend | 303,204 | (1/40) |
| antibody | Mouse monoclonal anti-human CD45RO-BV605 (UCHL1) | Biolegend | 562,790 | (1/40) |
| antibody | Mouse monoclonal anti-human CD20-BV786 (2H7) | Biolegend | 302,355 | (1/40) |
| antibody | Mouse monoclonal anti-human CD95-PE-Cy5 (DX2) | Becton Dickinson | 559,773 | (1/10) |
| antibody | Mouse monoclonal anti-human CD107a-PE-Cy5 (H4A3) | Becton Dickinson | 555,802 | (1/10) |
| antibody | Mouse monoclonal anti-human IFN-ɣ-AF700 (B27) | Life technologies | MHCIFG29 | (1/40) |
| antibody | Mouse monoclonal anti-human ULBP1-PerCP (170818) | R&D System | FAB1380C | (1/20) |
| antibody | Mouse monoclonal anti-human HLA-E-APC (3D12) | Biolegend | 342,606 | (1/20) |
| antibody | Mouse monoclonal anti-human CD155-BV786 (TX24) | Becton Dickinson | 744,720 | (1/100) |
| antibody | Mouse monoclonal anti-human MIC A/B–BV605 (6D4) | Becton Dickinson | 742,324 | (1/166) |
| antibody | Mouse anti-p24-PE (KC57) | Beckman Coulter | 6604667 | (1/200) |
| antibody | Mouse monoclonal anti-human CCR7-PE-CF594 (150503) | Becton Dickinson | 562,381 | (1/100) |
| antibody | Mouse monoclonal anti-human Ki67-BV510 (B56) | Becton Dickinson | 563,462 | (1/83) |
| recombinant DNA reagent | Plasmid encoding HIV-1 strain NL4.3 | NIH AIDS Reagent Program | NA | Malcom Martin |
| peptide, recombinant protein | BaL gp120 | NIH AIDS Reagent Program | NA | 1 µg |
| commercial assay or kit | Human PrimerFlow RNA Assay | EBioscience | NA | NA |
| commercial assay or kit | Human Fc block | Becton Dickinson | 564,219 | (1/20) |
| commercial assay or kit | mirVana miRNA isolation kit | Ambion | AM1560 | NA |
| commercial assay or kit | Zenon Human IgG labeling kit | Invitrogen | Z25451 | NA |
| chemical compound, drug | Ingenol | Sigma Aldrich | SML1318-1MG | 100 nM |
| chemical compound, drug | Romidepsin | Selleckchem | NA | 40 nM |
| chemical compound, drug | Ionomycine | Abcam, Inc. | ab120370 | 1 µM |
| chemical compound, drug | PMA | Abcam, Inc. | ab120297 | 81 nM |

*Continued on next page*

*Continued*

| Reagent type (species) or resource | Designation | Source or reference | Identifiers | Additional information |
|---|---|---|---|---|
| chemical compound, drug | Raltegravir | AIDS reagent program | NA | 1 µM |
| chemical compound, drug | Darunavir | AIDS reagent program | NA | 1 µM |
| chemical compound, drug | Nevirapine | Sigma Aldrich | SML0097-10MG | 1 µM |
| chemical compound, drug | Q-VD-OPh quinolyl-valyl-O-methylaspartyl-[–2,6-difluorophenoxy]-methyl ketone | Selleckchem | S7311 | 10 µM |
| chemical compound, drug | LIVE/DEAD Fixable Violet Dead Cell Stain Kit | Invitrogen | L34966 | (1/250) |
| software, algorithm | FlowJo software | TreeStar | NA | NA |

## Cells, virus, and reagents

PBMCs were obtained from PLWH and uninfected donors by Ficoll-Paque density gradient centrifugation and cryopreserved in liquid nitrogen. PBMCs were cultured in RPMI medium (Gibco) supplemented with 10% Fetal Bovine Serum (Gibco), 100 µg/ml streptomycin (Fisher Scientific), and 100 U/ml penicillin (Fisher Scientific) (R10 medium), and maintained at 37°C in a 5% $CO_2$ incubator. For RNA FISH-flow assays, fresh PBMCs were obtained from a whole blood donation (400 ml) from PLWH by Ficoll-Paque density gradient centrifugation and CD4+ T cells were immediately isolated and used without previous cryopreservation.

All plasmids needed for the generation of viral stocks and delta molecular clones were obtained through the NIH AIDS Reagent Program. Viral stocks were generated by transfection of 293T cells with the plasmids encoding the different molecular clones, and the resulting viral particles were titrated in TZMbl cells using an enzyme luminescence assay (britelite plus kit; PerkinElmer) as described previously (*Li et al., 2006*). BaL gp120 recombinant protein was obtained through the NIH AIDS Reagent Program. The A32 antibody was obtained through the AIDS Research and Reference Program, NIAID, NIH (Cat#11438) from Dr. James E. Robinson (*Moore et al., 1993*). Interleukin-2 (IL-2) was obtained from the Vall d´Hebron Hospital pharmacy. The pan-caspase inhibitor named Q-VD-OPh quinolyl-valyl-O-methylaspartyl-[–2,6-difluorophenoxy]-methyl ketone was purchased from Selleckchem.

## ADCC assay in cells coated with recombinant gp120

PBMCs from ART-suppressed PLWH, elite controllers, or uninfected donors were thawed and rested overnight in R10 medium. To exclude monocytes, PBMCs were cultured in a lying flask and adherent cells were discarded the next day. CD4+ T cells and NK cells were isolated from cryopreserved PBMCs using commercial kits (MagniSort Human CD4+ T Cell Enrichment; Affymetrix, and MagniSort Human NK cell Enrichment; eBioscience). Two rounds of cell separation were performed to maximize the purity of the cells (overall purity >85%). CD4+ T cells were stained with the membrane lipid marker PKH67 (Sigma-Aldrich) following the manufacturer's instructions, for the stable identification of these target cells, and then coated with 1 µg of recombinant gp120 protein for 1 hr at RT. A pool of uncoated cells was used as a negative control. After coating, target cells were extensively washed in ice-cold R10 medium and dispensed in U-bottom 96-well plates (100,000 cells per well, 10 wells per condition). After that, CD4+ T cells were incubated for 15 min with plasma (1:1,000 dilution) from a viremic (high viral load in blood) HIV+ patient. Then, NK effector cells were added at 1:1 target/effector ratio. Plates were centrifuged at 400xg for 3 min and incubated for 4 hr at 37°C and 5% $CO_2$. After incubation, cells were collected in FACs tubes, washed with staining buffer (PBS 3% FBS), and stained with anti-CCR7-PE-CF594 (150503, Becton Dickinson) for 30 min at 37°C. Next, cells were washed and stained with anti-CD3-AF700 (SK7, Biolegend), anti-CD45RO-BV605 (UCHL1, Biolegend), anti-CD32-PE-Cy7 (FUN-2, Biolegend), anti-HLA-DR-BV711 (L243, Biolegend), anti-CD20-BV786 (2H7, Biolegend), and anti-CD95-PE-Cy5 (DX2, Becton Dickinson) for 20 min at RT. Finally, cells were washed with staining buffer (PBS 3% FBS) and fixed with PFA (2%). Flow cytometry particles for absolute cell counting (5*10⁴/ml) (AccuCount Blank 5.0–5.9 µm, Cytognos) were added. Samples were acquired on an LSR

Fortessa flow cytometer (Becton Dickinson) and analyzed using FlowJo V10 software. We calculated the % of killing as the number of cells that disappeared in each population, following the next formula:

$$\% \text{ ADCC} = 100 - \frac{Cells\ (gp120)}{Cells\ (no\ gp120)} * 100$$

Control cells non-coated with gp120 (no gp120 condition) but incubated with plasma allowed us to properly measure ADCC responses and rule out a possible non-specific activation of the NK cells mediated by the plasma.

In some experiments, we assessed the capacity of NK cells to perform ADCC after being stimulated with IFN-α or IL-15 (Miltenyi Biotec). In such cases, PBMCs were stimulated overnight with 5000 U/ml IFN-α or 25 ng/ml of IL-15, and NK cells were isolated the next day.

## Assessment of the gp120-coating in the different CD4$^+$ T cell subsets

CD4$^+$ T cells were coated with recombinant gp120 as described above and then stained with anti-CCR7-PE-CF594 (150503, Becton Dickinson), anti-CD3-AF700 (SK7, Biolegend), anti-CD45RO-BV605 (UCHL1, Biolegend), anti-CD32-PE-Cy7 (FUN-2, Biolegend), anti-HLA-DR-BV711 (L243, Biolegend), anti-CD20-BV786 (2H7, Biolegend), and anti-CD95-PE-Cy5 (DX2, Becton Dickinson) antibodies. Next, cells were incubated for 20 min at RT with 5 µg/ml of A32 antibody (which binds to gp120 HIV protein). Then, cells were stained for 20 min at RT with an anti-human FITC-labelled secondary antibody (dilution 1:100) (Thermo Fisher) for A32 detection. Samples were acquired on an LSR Fortessa flow cytometer and data analyzed using FlowJo.

## Binding assessment of HIV-specific antibodies to T$_{CD32}$$^{dim}$ cells

Isolated CD4$^+$ T cells from uninfected donors were coated with gp120 recombinant protein. To detect if IC might result in steric hindrance precluding the binding of HIV-specific Igs to the T$_{CD32}$$^{dim}$ subset, we labeled the gp120-specific antibody (A32 mAb) with Allophycocyanin (APC) following the manufacturer instructions (Zenon Human IgG labeling kit, Invitrogen) and measured cell binding by flow cytometry. A pool of non-coated cells was used as a negative control. Cells were incubated with plasma containing high titer of non-HIV specific immune complexes (dilution 1000) for 20 min at RT. After, cells were incubated with A32-APC (3.3 µg/ml) for 25 min and labeled with anti-CD3-AF700 (SK7, Biolegend) and anti-CD32-PE-Cy7 (FUN-2, Biolegend). After, cells were washed once with PBS and stained with LIVE/DEAD Fixable Violet Viability (Invitrogen) for 20 min at RT. Finally, cells were washed with PBS and fixed with PFA (2%). Samples were acquired on an LSR Fortessa flow cytometer and data analyzed using FlowJo.

## ADCC-mediated NK cell activation of isolated T$_{CD32}$$^{dim}$ cells

For cell sorting experiments, 100 million PBMCs from healthy donors were stained with LIVE/DEAD violet viability (Invitrogen) for 20 min at RT. After washing, cells were surface stained with anti-CD3-PerCP (SK7; Becton Dickinson), anti-CD56-FITC (B159, Becton Dickinson), anti-CD32-PE-Cy7 (FUN-2, Biolegend), and anti-CD4-BV605 (RPA-T4, Becton Dickinson) antibodies for 20 min at RT. Cells were then washed and immediately sorted using a BD FACSAria Cell Sorter. We sorted the populations CD4$^-$CD3$^-$CD56$^+$ (NK cells), CD3$^+$CD4$^+$CD32$^+$ (T$_{CD32}$$^{dim}$) and CD3$^+$CD4$^+$CD32$^-$ (T$_{CD32}$$^-$). Then, we performed the ADCC assay and measured NK cell activation by flow cytometry. Purity of the cells was >97% in all cases.

Sorted T$_{CD32}$$^{dim}$ and T$_{CD32}$$^-$ cells were coated with recombinant gp120 as described above and incubated with plasma from a viremic HIV-infected patient at 1:1000 dilution 15 min before the addition of NK cells (ratio 1:2). Co-cultures were maintained for 4 hr in a 96-well plate at 37°C and 5% CO$_2$. NK cytotoxicity was assessed by measurement of CD107a and IFN-γ. As a positive control, we included NK cells cultured with 10 ng/ml PMA plus 1 µM ionomycin, and as a negative control, NK cells were cultured without any stimulus. CD107a-PE-Cy5 (H4A3; Beckton Dickinson), BD GolgiPlug Protein Transport Inhibitor (Becton Dickinson), and BD GolgiStop Protein Transport Inhibitor containing monensin (Becton Dickinson) were also added to each well at the recommended concentrations at the beginning of cell culture. After incubation, cells were washed and stained with a viability dye (LIVE/DEAD Fixable Violet dead cell stain; Invitrogen). Cells were then stained with anti-CD56-FITC (B159; Becton Dickinson), anti-CD32-PE-Cy7 (FUN-2, Biolegend) and, anti-CD4-BV605 (RPA-T4, Becton Dickinson)

antibodies for 20 min at RT. After that, cells were washed and fixed and permeabilized with Fixation/Permeabilization Solution (Becton Dickinson) for 20 min at 4°C, washed with BD Perm/Wash buffer, and stained with anti-IFN-γ AF700 (Life technologies) for 30 min at 4°C. After washing, cells were fixed with PFA (2%) and acquired on an LSR Fortessa flow cytometer (Becton Dickinson). Flow cytometry particles for absolute cell counting ($5*10^4$/ml) (AccuCount Blank 5.0–5.9 μm, Cytognos) were added. Data were analyzed using FlowJo V10 software.

## Ex vivo infection of unstimulated PBMCs

PBMCs from ART-suppressed PLWH or healthy donors were thawed and incubated overnight in R10 medium containing 40 U/ml IL-2. The next day, $CD4^+$ T cells were isolated using a commercial kit (MagniSort Human $CD4^+$ T Cell Enrichment; Affymetrix) and infected by incubation for 4 hr at 37°C with 156,250 or 2500 $TCID_{50}$ (50% tissue culture infectious dose) of $HIV_{NL4.3}$ or $HIV_{BaL}$ viral strains, respectively. In some experiments, cells were infected by spinoculation at 1200× $g$ for 2 hr at 37°C ($TCID_{50}$ of 78,125 and 625 for $HIV_{NL4.3}$ and $HIV_{BaL}$). Cells were then washed twice with PBS and cultured at 1 M/ml in a 96-well plate round-bottom with R10 containing 100 U/ml of IL-2 for the next 5 days. The resulting HIV-infected $CD4^+$ T cells were used for different experiments listed below.

## Phenotyping of HIV-infected $CD4^+$ T cells

HIV-infected cells were stained with LIVE/DEAD AQUA viability (Invitrogen) for 30 min at RT. After washing once with staining buffer, cells were stained with anti-ULBP1-PerCP (170818, R&D Systems), anti-CD32-PE-Cy7 (FUN-2, Biolegend), anti-CD3-PE-Cy5 (UCHT-1, Biolegend), anti-CD4-AF700 (RPA-T4, Becton Dickinson), anti-HLA-E-APC (3D12, Biolegend), anti-CD155-BV786 (TX24, Becton Dickinson) and anti-MIC A/B –BV605 (6D4, Becton Dickinson) antibodies for 20 min at RT. Cells were then fixed and permeabilized with Fixation/Permeabilization Solution (Becton Dickinson) for 20 min at 4°C, washed with BD Perm/Wash buffer, and stained with anti-p24-PE (Beckman Coulter) for 20 min on ice and 20 min at RT. After washing with BD Perm/Wash buffer, cells were fixed with PFA (2%). Samples were acquired on an LSR Fortessa flow cytometer and data analyzed using FlowJo V10 software. Gating was performed according to the different FMO controls.

## Assessment of the proliferative potential of $T_{CD32}^{dim}$ cells

HIV-infected cells were incubated with plasma from a viremic $HIV^+$ patient or plasma from an uninfected healthy donor (dilution 1:1000) for 4 hr at 37°C and 5% $CO_2$. Additionally, a pool of HIV-infected cells previously treated with an Fc receptor blocker (human Fc block, Becton Dickinson) was used as a control. After incubation, cells were washed and stained with LIVE/DEAD Fixable Violet Viability (Invitrogen) for 20 min at RT. Next, cells were washed with staining buffer and stained with anti-CCR7-PE-CF594 (150503, Becton Dickinson) for 30 min at 37°C. After washing, cells were stained with anti-CD56-FITC (B159, Becton Dickinson), anti-CD3-AF700 (SK7, Biolegend), anti-CD45RO-BV605 (UCHL1, Biolegend), anti-CD32-PE-Cy7 (FUN-2, Biolegend), anti-CD20-BV786 (2H7, Biolegend), anti-CD95-PE-Cy5 (DX2, Becton Dickinson), and anti-CD4-APC (OKT4, Biolegend) for 20 min at RT. Cells were then fixed and permeabilized with Fixation/Permeabilization Solution (Becton Dickinson) for 20 min at 4°C, washed with BD Perm/Wash buffer and stained with anti-p24-PE (KC57, Beckman Coulter) for 20 min on ice, and during the additional 30 min at RT an anti-Ki67-BV510 (B56, Becton Dickinson) was added. Finally, cells were washed and fixed with PFA (2%). Positive cells for the ki67 marker were determined according to FMO controls. Samples were acquired in an LSR Fortessa flow cytometer (Becton Dickinson) and analyzed with FlowJo V10 software.

## Natural cytotoxicity and ADCC NK-based assays

Five days after infection, HIV-infected cells were placed in a 96 round bottom well plate at 100,000 cells/well (ten replicates). Autologous NK cells, previously isolated by negative selection using magnetic beads (MagniSort Human NK cell Enrichment Kit, eBioscience), from PBMCs thawed the day before the co-culture, were added at 1:1 ratio. For the study of the ADCC response, plasma from a viremic HIV-infected patient containing a mix of antibodies targeting different HIV epitopes was added at 1:1000 dilution to HIV-infected cells 15 min before the addition of NK cells. After, the plate was centrifuged at 400× $g$ for 3 min to facilitate cell contact and then incubated at 37°C with 5% $CO_2$ for 4 hr. After, cells were collected in FACs tubes, washed with PBS, and stained with LIVE/DEAD

Fixable Violet Viability (Invitrogen) for 20 mins at RT. Next, cells were washed with staining buffer and stained first with anti-CCR7-PE-CF594 for 30 min at 37°C, and after an additional wash, with anti-CD56-FITC (B159, Becton Dickinson), anti-CD3-AF700 (SK7, Biolegend), anti-CD45RO-BV605 (UCHL1, Biolegend), anti-CD32-PE-Cy7 (FUN-2, Biolegend), anti-CD20-BV786 (2H7, Biolegend), anti-CD95-PE-Cy5 (DX2, Becton Dickinson), and anti-CD4-APC (OKT4, Biolegend) antibodies for 20 min at RT. Cells were then fixed and permeabilized with Fixation/Permeabilization Solution (Becton Dickinson) for 20 min at 4°C, washed with BD Perm/Wash buffer, and stained with anti-p24-PE (KC57, Beckman Coulter) for 20 min on ice and 20 min at RT. Finally, cells were washed and fixed with PFA (2%). Flow cytometry particles for absolute cell counting ($5*10^4$/ml) (AccuCount Blank 5.0–5.9 µm, Cytognos) were added. We also included a non-infected condition. Gating of the CD32 subpopulation was done according to FMO controls. Samples were acquired in an LSR Fortessa flow cytometer and analyzed with FlowJo software. The percentage of ADCC killing was determined by calculating the number of cells that disappeared in each infected population, normalizing numbers to the condition of HIV-infected CD4$^+$ T cells plus NK cells (in the absence of plasma). The NC response was quantified as the reduction in the % of p24 HIV protein in $T_{CD32}^-$ and $T_{CD32}^{dim}$ subsets, in comparison to the % of p24 in these subsets in basal conditions in the absence of NK cells. In these assays, cells were first stained with AQUA viability (Thermo Fisher), then with a panel of surface antibodies: anti-ULBP1-PerCP (170818, R&D Systems), anti-CD56-FITC (B159, Beckton Dickinson), anti-CD32-PE-Cy7 (FUN-2, Biolegend), anti-CD3-PE-Cy5 (UCHT-1, Biolegend), anti-CD4-AF700 (RPA-T4, Becton Dickinson), anti-HLA-E-APC (3D12, Biolegend), anti-CD155-BV786 (TX24, Becton Dickinson), anti-MIC A/B –BV605 (6D4, Becton Dickinson); and finally intracellularly stained with anti-p24-PE (KC57, Beckman Coulter) as previously described.

Cell conjugates were identified by flow cytometry and quantified as follows: live lymphocytes (isolated CD4$^+$ T and NK cells) were gated after excluding contamination with B cells and monocytes. Then, HIV-infected cells were identified as p24$^+$ cells in CD3$^+$ cells, from which we selected separately both, CD32$^{dim}$ and CD32$^{neg}$ cells. From CD32$^{dim}$ and CD32$^{neg}$ populations we gated cell doublets by side scatter signals, and then we determined the fraction of cell doublets composed by CD4$^+$ T cells (CD3$^+$ and CD32$^{dim}$ or CD32$^{neg}$) and NK cells (identified as CD56$^+$ cells).

## Detection of cells expressing HIV-1 RNA by the RNA FISH-flow assay

PBMCs from nine ART-treated PLWH were obtained from a whole blood donation (400 ml) and CD4$^+$ T cells were isolated by a negative selection kit (MagniSort Human CD4$^+$ T Cell Enrichment; eBioscience). At least $6 \times 10^6$ of freshly-isolated CD4$^+$ T cells were studied per condition, being subjected to viral reactivation with different LRAs (Ingenol and Romidepsin), and including the positive (PMA plus ionomycin) and negative controls (medium condition). Before viral reactivation, cells were preincubated with the pan-caspase inhibitor Q-VD-OPh (Selleckchem) for 2 hr. In addition, to prevent new rounds of viral infection during HIV reactivation, the cells were treated with LRAs in the presence of Raltegravir (1 µM) for 22 hr. Afterward, the RNA FISH-flow assay was performed according to the manufacturer's instructions (Human PrimerFlow RNA Assay, eBioscience) with some modifications as previously described (*Grau-Expósito et al., 2017*). Briefly, after antibody staining and cell fixation and permeabilization, cells were ready for hybridization with a set of 50 probes spanning the whole Gag-Pol HIV mRNA sequence (bases 1165–4402 of the HXB2 consensus genome). Next, the cells were subjected to amplification signal steps, and HIV RNA was detected using Alexa Fluor 647-labeled probes. In these experiments, for surface antigen labeling, anti-human CD3 (AF700, Biolegend), anti-human CD32 (PE-Cy7, Biolegend), anti-human CD20 (BV785, Biolegend), and anti-human HLA-DR (BV711, Biolegend) antibodies were used, and cell viability analyzed with violet viability dye (Invitrogen). Samples were analyzed with the LSR Fortessa flow cytometer, and the results were analyzed with FlowJo v10 software.

## Ex vivo viral reactivation of the natural reservoir from ART-suppressed PLWH

CD4$^+$ T lymphocytes from ART-suppressed PLWH were isolated as described above and cultured in R10 medium with Q-VD-OPh (Selleckchem) for 2 hr in the presence of Raltegravir (1 µM), Darunavir (1 µM), and Nevirapine (1 µM) to prevent new rounds of viral infection. After 2 hr, PMA plus ionomycin (PMA 81 nM; ionomycin 1 µM) were added to the cell culture as a latency reversal agent and left for

18 hr to reactivate latent HIV. After viral reactivation, cells were subjected to cytotoxicity and ADCC assays as previously described here. For these experiments, we used the autologous plasma from each patient. After, cells were stained with LIVE/DEAD Far Red viability for 20 min at RT and then with anti-CD32 (FITC, Biolegend) and anti-CD3 (PerCP, Becton Dickinson) antibodies for 20 min at RT. After the cell surface staining, cells were fixed and permeabilized with Fixation/Permeabilization Solution (Becton Dickinson) for 20 min at 4°C, washed with BD Perm/Wash buffer, and stained with anti-p24-PE (Beckman Coulter) for 20 min on ice and 20 min at RT. Finally, cells were washed and fixed with PFA (2%). Samples were acquired in a FACSCalibur flow cytometer (Becton Dickinson) and analyzed with FlowJo V10 software. The limit of detection of the assay was established at 50 p24$^+$ cells/million CD4$^+$ T cells (*Figure 2—figure supplement 1C*).

## Quantification of HIV DNA and HIV RNA by quantitative PCR

CD4$^+$ T lymphocytes were enriched from total PBMCs using a negative selection kit (MagniSort Human CD4$^+$ T Cell Enrichment, eBioscience). Then, CD4$^+$ T cells were fractioned to extract RNA or quantify the HIV DNA from cell lysates. CD4$^+$ T cells for DNA analysis were immediately lysed with proteinase K-containing lysis buffer (at 55°C overnight and 95°C for 5 min). The HIV DNA in the cell lysates was quantified by qPCR using primers and probes specific for the 1-LTR HIV region (LTR forward 5′-TTAAGCCTCAATAAAGCTTGCC-3′, LTR reverse 5′-GTTCGGGCGCCACTGCTAG-3′, and probe 5′ /56-FAM/CCAGAGTCA/ZEN/CACAACAGACGGGCA/31ABkFQ/3′). CCR5 gene was used for cell input normalization. Samples were analyzed in an Applied Biosystems 7000 Real-Time PCR System. For Viral RNA quantification, CD4$^+$ T cells were subjected to RNA extraction using the mirVana kit following the manufacturer's instructions (Ambion). Reverse transcription of RNA to cDNA was performed with SuperScriptIII (Invitrogen), and cDNA was quantified by qPCR with primers against the HIV long terminal repeat (LTR). Quantification of RNA and DNA copies was performed using a standard curve, and values were normalized to 1 million CD4$^+$ T cells.

## Statistical analyses

Statistical analyses were performed with Prism software, version 6.0 (GraphPad). A $p<0.05$ was considered significant.

## Acknowledgements

This study was supported by the Spanish Secretariat of Science and Innovation and FEDER funds (grants SAF2015-67334-R and RTI2018-101082-B-I00 [MINECO/FEDER]), the Spanish "Ministerio de Economia y Competitividad, Instituto de Salud Carlos III" (ISCIII, PI17/01470), GeSIDA and the Spanish AIDS network Red Temática Cooperativa de Investigación en SIDA (RD16/0025/0007), the Fundació La Marató TV3 (grants 201805-10FMTV3 and 201814-10FMTV3) and the Gilead fellowships GLD19/00084 and GLD18/00008. M.B is supported by the Miguel Servet program funded by the Spanish Health Institute Carlos III (CP17/00179). A.A-G is supported by the Spanish Secretariat of Science and Innovation Ph.D. fellowship (BES-2016–076382). The funders had no role in study design, data collection, and analysis, the decision to publish, or preparation of the manuscript.

## Additional information

### Funding

| Funder | Grant reference number | Author |
| --- | --- | --- |
| Spanish National Plan for Scientific and Technical Research and Innovation | SAF2015-67334-R | Maria J Buzon |
| Spanish National Plan for Scientific and Technical Research and Innovation | RTI2018-101082-B | Maria J Buzon |
| Fundació La Marató TV3 | 201805-10FMTV3 | Maria J Buzon |

| Funder | Grant reference number | Author |
|---|---|---|
| Fundació La Marató TV3 | 201814-10FMTV3 | Meritxell Genescà |
| Spanish Health Institute Carlos III | CP17/00179 | Maria J Buzon |
| Spanish National Plan for Scientific and Technical Research and Innovation | BES-2016-076382 | Antonio Astorga-Gamaza |

The funders had no role in study design, data collection and interpretation, or the decision to submit the work for publication.

## Author contributions

Antonio Astorga-Gamaza, Conceptualization, Data curation, Formal analysis, Methodology, Validation, Visualization, Writing – original draft; Judith Grau-Expósito, Data curation, Formal analysis, Methodology, Writing – review and editing; Joaquín Burgos, Jordi Navarro, Adrià Curran, Bibiana Planas, Paula Suanzes, Vicenç Falcó, Investigation, Resources, Writing – review and editing; Meritxell Genescà, Conceptualization, Investigation, Methodology, Writing – review and editing; Maria J Buzon, Conceptualization, Data curation, Formal analysis, Funding acquisition, Investigation, Methodology, Project administration, Supervision, Visualization, Writing – original draft, Writing – review and editing

## Author ORCIDs

Jordi Navarro http://orcid.org/0000-0002-7187-0367
Paula Suanzes http://orcid.org/0000-0002-6871-0098
Maria J Buzon http://orcid.org/0000-0003-4427-9413

## Ethics

Human subjects: This study involves human samples. PBMCs from PLWH were obtained from the HIV unit of the Hospital Universitari Vall d'Hebron in Barcelona, Spain. Study protocols were approved by the corresponding Ethical Committees (Institutional Review Board numbers PR(AG)270/2015 and PR(AG)39/2016). PBMCs from healthy donors were obtained from the Blood and Tissue Bank, Barcelona, Spain. All subjects recruited to this study were adults who provided written informed consent. Samples were completely anonymous and untraceable and were prospectively collected and cryopreserved in the Biobank (register number C.0003590).

## Decision letter and Author response

Decision letter https://doi.org/10.7554/eLife.78294.sa1
Author response https://doi.org/10.7554/eLife.78294.sa2

# Additional files

## Supplementary files

• Supplementary file 1. Clinical data of PLWH included in the study.
• MDAR checklist

## Data availability

The authors declare that the data supporting the findings of this study are available within the paper and its supplementary information files. Source data are provided with this paper.

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
