## [Editor Report]

Persistence of the viral reservoir is hampering HIV cure. This study describes a possible way that HIV-infected cells in the reservoir may escape antibody killing. The findings show that reservoir cells tend to have less of a receptor that binds HIV antibodies capable of cell killing – these cells may then have a survival advantage as they are less susceptible to antibody killing. The study suggests that they also seem to be susceptible to proliferation, which helps maintain the reservoir. These studies provide evidence for one way in which the HIV reservoir is maintained.

---

## [Decision Letter]

**Decision letter after peer review:**

Thank you for submitting your article "Identification of HIV-Reservoir Cells with Reduced Susceptibility to Antibody-Dependent Immune Response" for consideration by *eLife*. Your article has been reviewed by 3 peer reviewers, one of whom is a member of our Board of Reviewing Editors, and the evaluation has been overseen by Satyajit Rath as the Senior Editor. The following individual involved in review of your submission has agreed to reveal their identity: Nilu Goonetilleke (Reviewer #2).

Essential revisions:

1) The HLA-E data is not convincing. It is important that they experimentally resolve whether the effect the results observed reflect HLA-E blocking or just mimic the isotype control. If they can't do this, then I think they'd should remove Figure 6. The more cautious interpretation of the current data would just be its a non-specific effect presumably Fc effect of Ab.

– The HLA-E blocking experiments are not convincing. HLA-E is hard to block. Data to demonstrate effective blocking is needed. This is particularly important given the authors observe the same trend with their Isotype control Ab.

– The fact that an isotype control was able to increase ADCC similar to an anti-HLA-E mAb (Figure 6 D and E) is confusing. Do the authors have a hypothesis for this observation, other than to say the increase is non-anti-HLA-E-specific?

– Figure 6 as noted above, please provide data on HLA-E blocking. I would note, that if the authors cannot show this convincing, removal of these data from manuscript would, to me, not be fatal. The authors show convincing data that HLA-E is elevated on CD32 lo cells and HLA-E /NK interactions are well established.

2) There are many comparisons presented and the authors should do some analyses that correct for multiple comparison and comment on those results and what remains significant.

---

## [Author Response]

Essential revisions:1) The HLA-E data is not convincing. It is important that they experimentally resolve whether the effect the results observed reflect HLA-E blocking or just mimic the isotype control. If they can't do this, then I think they'd should remove Figure 6. The more cautious interpretation of the current data would just be its a non-specific effect presumably Fc effect of Ab.– The HLA-E blocking experiments are not convincing. HLA-E is hard to block. Data to demonstrate effective blocking is needed. This is particularly important given the authors observe the same trend with their Isotype control Ab.– The fact that an isotype control was able to increase ADCC similar to an anti-HLA-E mAb (Figure 6 D and E) is confusing. Do the authors have a hypothesis for this observation, other than to say the increase is non-anti-HLA-E-specific?– Figure 6 as noted above, please provide data on HLA-E blocking. I would note, that if the authors cannot show this convincing, removal of these data from manuscript would, to me, not be fatal. The authors show convincing data that HLA-E is elevated on CD32 lo cells and HLA-E /NK interactions are well established.

Our goal here was to test if the upregulation of HLA-E on infected CD32+ cells played a role in evading the NK effector response. Similar experimental settings (the use of an anti-HLA-E Ab to block the HLA-E-NKG2A axis) have been used before by other investigators (i.e https://doi.org/10.1073/pnas.95.9.5199). We described in the manuscript that the result was not specific, and it was more likely mediated by the Fc portion of the antibody. However, we believe that there are alternative intrinsic factors in CD32+ cells that promote their resistance to NK killing. Thus, we agree with the reviewers that the results are not conclusive, and removing the data does not impact the main conclusions of our study. Then, we have removed the HLA-E blocking data in Figure 6.

2) There are many comparisons presented and the authors should do some analyses that correct for multiple comparison and comment on those results and what remains significant.

As suggested by the reviewers, we have performed the statistical tests corrected for multiple comparisons, when required. Specifically, we have run the test in Figures 1A, B, and C; Figure 2A; Figure 3 I-P; Figure 5C; Figure 6A. The name of the statistical tests applied is depicted in each Figure legend. Overall, main statistically significant comparisons are maintained with the new analyses.